# The AP2/ERF GmERF113 Positively Regulates the Drought Response by Activating *GmPR10-1* in Soybean

**DOI:** 10.3390/ijms23158159

**Published:** 2022-07-24

**Authors:** Xin Fang, Jia Ma, Fengcai Guo, Dongyue Qi, Ming Zhao, Chuanzhong Zhang, Le Wang, Bo Song, Shanshan Liu, Shengfu He, Yaguang Liu, Junjiang Wu, Pengfei Xu, Shuzhen Zhang

**Affiliations:** 1Soybean Research Institute of Northeast Agricultural University/Key Laboratory of Soybean Biology of Chinese Education Ministry, Harbin 150030, China; fangxin0622@163.com (X.F.); majia15504500812@163.com (J.M.); guofengcai2022@163.com (F.G.); qidongyue1112@163.com (D.Q.); 13149511790@163.com (M.Z.); zhangchuanzhong92@126.com (C.Z.); wl18645003532@163.com (L.W.); songbo99@neau.edu.cn (B.S.); ars336699@aliyun.com (S.L.); shengfuhe1996@163.com (S.H.); liuyaguang929@163.com (Y.L.); 2Key Laboratory of Soybean Molecular Design Breeding, Northeast Institute of Geography and Agroecology, Chinese Academy of Sciences, Harbin 150030, China; 3Soybean Research Institute of Heilongjiang Academy of Agricultural Sciences/Key Laboratory of Soybean Cultivation of Ministry of Agriculture, Harbin 150030, China; nkywujj@126.com

**Keywords:** soybean, drought tolerance, GmERF113, *GmPR10-1*, ABA

## Abstract

Ethylene response factors (ERFs) are involved in biotic and abiotic stress; however, the drought resistance mechanisms of many ERFs in soybeans have not been resolved. Previously, we proved that *GmERF113* enhances resistance to the pathogen *Phytophthora sojae* in soybean. Here, we determined that *GmERF113* is induced by 20% PEG-6000. Compared to the wild-type plants, soybean plants overexpressing *GmERF113* (*GmERF113*-OE) displayed increased drought tolerance which was characterized by milder leaf wilting, less water loss from detached leaves, smaller stomatal aperture, lower Malondialdehyde (MDA) content, increased proline accumulation, and higher Superoxide dismutase (SOD) and Peroxidase (POD) activities under drought stress, whereas plants with *GmERF113* silenced through RNA interference were the opposite. Chromatin immunoprecipitation and dual effector-reporter assays showed that GmERF113 binds to the GCC-box in the *GmPR10-1* promoter, activating *GmPR10-1* expression directly. Overexpressing *GmPR10-1* improved drought resistance in the composite soybean plants with transgenic hairy roots. RNA-seq analysis revealed that GmERF113 downregulates abscisic acid 8′-hydroxylase 3 (*GmABA8*’*-OH 3*) and upregulates various drought-related genes. Overexpressing *GmERF113* and *GmPR10-1* increased the abscisic acid (ABA) content and reduced the expression of *GmABA8*’*-OH3* in transgenic soybean plants and hairy roots, respectively. These results reveal that the GmERF113-GmPR10-1 pathway improves drought resistance and affects the ABA content in soybean, providing a theoretical basis for the molecular breeding of drought-tolerant soybean.

## 1. Introduction

Drought is a major abiotic stress that hampers the growth, development, and productivity of plants. In the context of rising global population and climate change, freshwater resources are becoming increasingly insufficient to meet agricultural needs [1]. In plants, drought leads to a series of changes at the morphological, physiological, and molecular levels. For example, the stomatal aperture is reduced to maintain the plant body moisture, and root growth is promoted to enhance the plant’s capacity to absorb water [2,3]. The activity of antioxidant enzymes such as superoxide dismutase (SOD), peroxidase (POD), catalase, and ascorbate peroxidase (APX) and the contents of osmolytes such as proline increase to help maintain the overall plant homeostasis [4,5,6]. The contents of phytohormones are altered, for example through increased abscisic acid (ABA) accumulation, to regulate stomatal movement, which in turn helps the plants tolerate the effects of extreme water deficit [7,8]. Numerous genes that are related to various signaling pathways, metabolic pathways, and functional proteins are induced to prevent or mitigate the damage that is caused by drought stress; these include genes that are related to ABA synthesis (e.g., *NCED5*) and catabolism (e.g., *CYP707A*) and to cell wall components (e.g., *β-Galactosidase 10*), late embryogenesis abundant (LEA) genes, and heat shock protein (HSP) genes [9,10,11,12].

Transcription factors such as bZIP, NAC, MYB, WRKY, and AP2/ERF family members play key regulatory roles in transcriptional reprogramming, thereby coordinately regulating the expression of many defense-related genes [13,14,15,16,17,18,19]. The AP2/ERF superfamily of plant-specific transcription factors (TFs) are characterized by a 60- to 70-amino-acid conserved AP2/ERF domain [20,21]. The AP2/ERF transcription factor superfamily, which has been identified in various species, is divided into different families based on the number of AP2 domains, including the AP2, ERF, DREB, and RAV families [20,21,22,23,24,25,26]. 

There is growing evidence that ethylene response factor (ERF) subfamily members are involved in plant responses and adaptation to abiotic stress. For example, overexpressing the AP2/ERF family gene *BpERF13* in birch (*Betula platyphylla*) enhanced cold tolerance by upregulating *CBF* genes and mitigating reactive oxygen species (ROS)-induced damage [27]. Overexpressing the ethylene response factor gene *ERF96* enhanced selenium tolerance in *Arabidopsis thaliana* [28]. ERF genes from maize (*Zea mays* L.; *ZmERF*), alfalfa (*Medicago sativa* L.; *MsERF*), Arabidopsis (*Arabidopsis thaliana* L.; *AtERF1*), peanut (*Arachis hypogaea* L.; *AhERF019*), wheat (*Triticum aestivum* L.; *TaERF3*), and soybean (*Glycine max* (L). Merr.; *GmERF5*) are all induced by drought [29,30,31,32,33,34,35]. Overexpressing *OsERF71* and *AtERF019* protected rice (*Oryza sativa*) and Arabidopsis from drought by altering the structure of rice roots, Arabidopsis stomatal aperture and cell wall permeability, respectively [11,36]. *TaERF3-*overexpressing wheat plants and *OsLG3-*overexpressing rice plants increased drought resistance by accumulating proline, chlorophyll, and increasing SOD and POD activities, respectively [32,37]. OsLG3 and OsERF71 also participated in the drought tolerance of rice by regulating the expression of ROS-scavenging and ABA-signaling-related genes [34,37], and the ERF protein MdERF38 responded to drought stress by interacting with a positive modulator of anthocyanin biosynthesis (MdMYB1) and facilitating its binding to its target genes in apple (*Malus domestica*) [38].

The AP2/ERF domain is required for the activity of AP2/ERF TFs, which it triggers by binding to cis-acting elements in their target genes, such as GCC-box motifs, DRE/CRT, or the TTG motifs in their promoters [32,39,40,41]. AP2/ERF TFs bind to the DRE/CRT or GCC-box motifs in stress-responsive genes to enhance resistance to abiotic stress [42,43]. For example, the AP2 transcription factor NtERF172 confers drought resistance in tobacco (*Nicotiana tabacum*) by binding to the DRE motif in the *NtCAT* promoter [44]. Overexpressing *AtERF53* improved the drought tolerance in Arabidopsis by binding to the GCC-boxes and/or DRE elements in the promoters of its downstream genes, such as *COR15B* and *P5CS1* [45].

Pathogenesis-related (PR) genes are known to contribute to plant resistance to biotic stresses [46,47,48,49,50,51,52,53]. However, there are also numerous studies showing that PR genes also confer tolerance to abiotic stresses. For example, ectopic overexpression of a salt-stress-induced PR Class 10 protein (PR10) gene from peanut afforded broad-spectrum abiotic stress tolerance in transgenic tobacco (*Nicotiana tabacum*) [54]. *AhSIPR10* from peanut (*Arachis hypogaea*), *RSOsPR10* from rice (*Oryza sativa*), and *ThPR10* from *Tamarix hispida* are induced by drought and salt stress [54,55,56,57]. In rice, overexpressing the root-specific PR gene *RSOsPR10* strongly improved drought tolerance [55] and overexpressing *JIOsPR10* increased drought and salt stress tolerance [48]. SlNPR1 enhances the drought tolerance in tomato (*Solanum lycopersicum*) by regulating key drought-related genes, including *SlGST*, *SlDHN*, and *SlDREB* [58]. *PR1-*, *PR2-*, and *PR5*-overexpressing Arabidopsis plants showed enhanced drought tolerance. Finally, the transcription factor Di19 (drought-induced 19) directly upregulates *PR1*, *PR2*, and *PR5* in response to drought stress in Arabidopsis [59].

To date, a total of 160 ERF genes have been identified in soybean [60], but the functions of most of these genes have not been resolved. We previously demonstrated that the ERF gene *GmERF113* (GenBank accession no. XM_003548806, NCBI protein no. XP_003548854) could bind to the GCC-box and overexpression of *GmERF113* transgenic soybean plants enhanced the expression of *GmPR10-1* and resistance to *P. sojae* [61]. The objectives of this study were (i) to investigate whether *GmERF113* is involved in drought stress through expression analysis; (ii) to verify the drought resistance function of *GmERF113* through the phenotype and the determination of drought-related physiological indicators; and (iii) to resolve a molecular mechanism of GmERF113 in regulating drought resistance in soybean: the GmERF113-GmPR10-1 pathway enhances drought resistance and affects the ABA content in soybean, which will help us better understand the drought resistance function of GmERF113 and provide a theoretical basis for the molecular breeding of drought-tolerant soybean.

## 2. Results

### 2.1. GmERF113 Is Induced by Drought

To investigate whether *GmERF113* is associated with drought stress, we performed quantitative reverse-transcription PCR to detect the expression of *GmERF113* of soybean seedlings at the V2 stage that were treated with 20% PEG-6000 to induce drought (osmotic) stress. Under these PEG stress conditions, the transcript level of *GmERF113* increased rapidly from 1 to 9 h (peak level 21.86-fold of the control) and then decreased to a low level by 24 h (Figure 1). These results suggest that *GmERF113* is likely involved in the drought response in soybean.

### 2.2. Overexpression of GmERF113 Enhances Drought Tolerance in Transgenic Soybean Plants

To explore the role of *GmERF113* in drought tolerance, we generated the overexpression vector 35S:*GmERF113* and the RNA interference vector *GmERF113*-RNAi and used them to transform soybean plants via *Agrobacterium*-mediated transformation. A total of three independent *GmERF113*-OE T_3_ lines (*GmERF113*-OE15, *GmERF113*-25, and *GmERF113*-47) and three independent *GmERF113*-RNAi T_3_ lines (*GmERF113-*RNAi15, *GmERF113-*RNAi17, and *GmERF113-*RNAi21) were identified using bar strips and qRT-PCR (Appendix A). Immunoblot analysis further confirmed that *GmERF113* had been successfully introduced into and expressed in the *GmERF113*-OE soybean plants (Appendix A).

Since plant water retention capacity contributes to drought resistance, we measured the fresh weights of the detached second trifoliate leaves of the wild-type (WT), *GmERF113*-OE, and *GmERF113*-RNAi T_3_ plants over 4 h following their detachment from the plants. Compared to that in the WT plants, water loss was slower in the leaves from *GmERF113*-OE plants and faster in the leaves from *GmERF113*-RNAi plants (Figure 2A). These results indicate that overexpressing *GmERF113* enhances the water retention capacity of plants, thus improving their drought resistance.

To further validate the role of *GmERF113* in drought response, we assessed the drought tolerance of the WT, *GmERF131*-OE, and *GmERF113*-RNAi T_3_ plants that were grown in soil. When the second trifoliate leaves of the WT, *GmERF113*-OE, and *GmERF131*-RNAi T_3_ plants were fully expanded (until V3 stage), we withheld watering for 7 days and resumed watering for 2 days. There was no difference in the phenotypes of the WT, *GmERF113*-OE, and *GmERF113-*RNAi plants under well-watered conditions. However, after 5 days without watering, most of the leaves of *GmERF113*-RNAi plants started to wilt and crumple, and the lowermost leaves of the WT and *GmERF113*-OE plants started to turn yellow and wilt, but the extent of leaf wilting was greater in the WT plants than in *GmERF113*-OE plants. After withholding watering for 7 days, all the leaves of *GmERF113*-RNAi plants were wilted and brittle, and all the leaves of the WT plants were wilted, while most of the leaves of *GmERF113*-OE plants were still expanded. After 2 days of re-watering, the WT and *GmERF113*-RNAi plants did not recover and were dead, whereas *GmERF113*-OE plants returned to normal growth (Figure 2B).

We also observed the stomata in the second trifoliate leaves of WT, *GmERF113*-OE, and *GmERF113*-RNAi plants during drought treatment. Before drought treatment, the stomatal aperture of *GmERF113*-OE plants was significantly smaller than that of the WT and *GmERF113*-RNAi plants (*** P* < 0.01) (Figure 2C,D). After 5 days of drought treatment, the stomatal apertures of *GmERF113*-OE and *GmERF113*-RNAi plants were significantly smaller than those of the WT plants (*** P* < 0.01). At 7 days of drought treatment, the stomata of WT, *GmERF113*-OE, and *GmERF113*-RNAi plants were all closed. After 2 days of rehydration, only the stomata of *GmERF113*-OE plants were open, while the stomata of WT and *GmERF113*-RNAi plants failed to open (Figure 2C). Together, these results suggest that GmERF113 plays a positive role in regulating the drought response.

### 2.3. Overexpressing GmERF113 Alters the Physiological and Biochemical Parameters of Transgenic Soybean Plants under Drought Stress

SOD, POD, and proline help to protect plants from oxidative damage under stress, and the MDA content reflects the extent of membrane lipid peroxidation [62,63,64,65,66,67]. Therefore, we evaluated these parameters to examine the adaptation of soybean plants to drought stress. Under drought treatment, the MDA content was significantly lower in the *GmERF113*-OE transgenic plants than in the WT plants, whereas the MDA content in the *GmERF113-*RNAi plants remained high and was significantly higher than that of the WT plants (Figure 3A). These results indicate that *GmERF113*-OE transgenic plants suffered less oxidative damage than the WT and *GmERF113*-RNAi plants.

Proline is an important osmolyte that reduces the osmotic potential and alleviates dehydration stress in plants. The proline content of *GmERF113*-OE plants was more than 1.4-fold higher than that of the WT plants after 5 and 7 days of drought treatment, whereas the proline content of *GmERF113*-RNAi plants was approximately 26% and 17% lower than that of the WT plants after 5 and 7 days of drought treatment, respectively (Figure 3B).

SOD and POD scavenge superoxide radicals. Therefore, the SOD and POD activity can be used to analyze the drought stress response of plants. Under normal conditions, the SOD and POD activities of WT and transgenic lines were almost the same. Under drought stress conditions, however, the SOD and POD activities were significantly higher in *GmERF113*-OE than in the WT or *GmERF113*-RNAi plants (Figure 3C,D). These results suggest that overexpressing *GmERF113* affects the accumulation of stress-related compounds such as proline and MDA, as well as the activities of stress-related enzymes such as SOD and POD, thereby improving drought tolerance.

### 2.4. GmERF113 Directly Activates the Expression of GmPR10-1

The accumulation of PR proteins during microbial infection or under abiotic stress conditions constitute a component of the plant innate immune response [68]. A GCC-box cis-element is present in the promoters of many PR genes. GmERF113 was previously shown to bind to the GCC-box and overexpression of *GmERF113* transgenic soybean plants enhanced the expression of *GmPR10-1* and resistance to *P. sojae* [61]. Here, we found that the *GmPR10* promoter contains a GCC-box. To investigate whether GmERF113 directly binds to the GCC-box in the *GmPR10* promoter, we performed ChIP assays with *ERF1-myc* seedlings at the V3 stage. The −295/−158 fragment (containing the GCCGCC site at positions −170 to −164) of the *GmPR10-1* promoter in immunoprecipitated DNA from *GmERF113-myc* plants was significantly enriched (8.39-fold) compared with its abundance in the WT plants (Figure 4A,B).To further reveal the role of GmERF113 in regulating *GmPR10-1* expression, we cloned the full-length *GmPR10-1* promoter sequence (2068 bp) and fused it into the pGreenII 0800-LUC vector to generate the reporter construct *p35S: REN-pGmPR10-1: LUC* (Figure 4C). The recombinant vector *p35S:GmERF113-myc*, in which *GmERF113* expression is driven by the *35S* promoter, was used as the effector construct (Figure 4C). The reporter construct *p35S: REN-pGmPR10-1: LUC* and the *p35S* null effector vector, or the reporter construct *p35S: REN-pGmPR10-1:LUC* and the effector construct *p35S:GmERF113-myc*, were co-transfected into *N. benthamiana* leaves. After 3 days of co-transfection, the appropriate amounts of 1 mM D-luciferin were sprayed onto the transfected *N. benthamiana* leaves. The chemiluminescence signal in *N. benthamiana* leaves that were co-transfected with the reporter construct *p35S: REN-pGmPR10-1:LUC* and the effector construct *p35S:GmERF113-myc* was significantly stronger than that in the leaves harboring the reporter construct *p35S:REN-pGmPR10-1:LUC* and the effector construct *p35S* (Figure 4D). The results of the LUC/REN relative activity assay further indicated that GmERF113 directly promotes the expression of *GmPR10-1* (Figure 4E). These results imply that GmERF113 binds to the GCC-box in the *GmPR10-1* promoter and directly activates its expression.

### 2.5. Overexpression of GmPR10-1 Improves Drought Tolerance in Soybean Hairy Roots

To further investigate the role of *GmPR10-1* in drought response, we generated *GmPR10-1* transgenic soybean hairy roots via *Agrobacterium rhizogenes*-mediated transformation. GUS staining and qRT-PCR analysis confirmed the generation of transgenic hairy roots that were overexpressing *GmPR10-1* (Appendix A). PCR analysis confirmed that the recombinant *GmPR10-1* RNAi vector had been successfully transferred into the hairy roots, and qRT-PCR analysis showed that *GmPR10-1* expression in *GmPR10-1* RNAi transgenic hairy roots was successfully disrupted (Appendix A).

We selected empty vector (EV) control, *GmPR10-1*-OE, and *GmPR10-*1-RNAi transgenic soybean composite plants with hairy roots of the same length, transferred them to fresh vermiculite, incubated them for 3 days under normal watering conditions, and then withheld watering. After 5 days of drought treatment, all the leaves of the *GmPR10-1* RNAi transgenic composite plants were severely wilted, whereas the second and third trifoliate leaves of the EV composite plants and all the leaves of the *GmPR10-1-* OE transgenic composite plants showed no obvious wilting. After 7 days of drought treatment, the EV composite plants showed severe leaf dehydration and the *GmPR10-1* RNAi composite plants showed more severe leaf abscission compared to the *GmPR10-1-* OE composite plants. We then resumed watering, and 2 days after the start of rehydration, only the *GmPR10-1*-OE composite plants regained vitality, whereas both the EV and *GmPR10-1*-RNAi composite plants were dead (Figure 5B).

We then measured the water loss rate and the stomatal aperture of the second trifoliate leaves from the top in EV, *GmPR10-1*-OE, and *GmPR10-1*-RNAi composite soybean plants with transgenic hairy roots. Compared to that in the EV control plants, the rate of water loss rate from the leaves was reduced in the leaves of the *GmPR10-1*-OE plants and elevated in the leaves of the *GmPR10-1*-RNAi plants (Figure 5A). Before the drought treatment, the stomatal apertures of the *GmPR10-1*-OE composite plants were significantly smaller than those of the EV and *GmPR10-1*-RNAi plants (*** P* < 0.01). After 5 days of drought treatment, the stomatal apertures of *GmPR10-1*-OE and *GmPR10-1*-RNAi plants (*** P* < 0.01) were significantly smaller than those of the EV control plants. After 7 days of drought treatment, the stomata of the EV, *GmPR10-1*-OE, and *GmPR10-1*-RNAi plants were all closed. After 2 days of rehydration, only the stomata of *GmPR10-1*-OE composite plants were open, whereas the EV and *GmPR10-1*-RNAi stomata failed to open (Figure 5C,D). These results suggest that overexpressing *GmPR10-1* enhances drought tolerance in transgenic soybean composite plants with transgenic hairy root.

To further verify the role of *GmPR10-1* in drought resistance, we measured the changes in the physiological and biochemical parameters of the leaves of composite plants. Before drought treatment, there were no significant differences in the MDA and proline contents or SOD and POD activities in the leaves of EV, *GmPR10-1*-OE, and *GmPR10-1*-RNAi composite plants. After 5 days and 7 days of drought treatment, compared to the EV and *GmPR10-1*-RNAi composite plants, the MDA content (Figure 6A) in the leaves of *GmPR10-1*-OE composite plants was significantly reduced, while the proline content and the SOD and POD activities were significantly increased (Figure 6B–D). Together, these results demonstrate that overexpressing *GmPR10-1* improves the drought tolerance of soybean composite plants.

### 2.6. GmERF113 Differentially Regulates Genes Involved in Drought Response

To further investigate the drought resistance mechanism of *GmERF113*, we performed comparative transcriptome analysis using leaves from plants of three *GmERF113*-OE lines and WT plants. Based on the RNA-seq data, we identified 360 differential expression genes (DEGs) in the *GmERF113*-OE plants compared to the control, including 57 upregulated and 303 downregulated genes (Figure 7A,B). To examine the functions of these DEGs, we performed GO analysis. All of the DEGs were assigned to three major categories: biological processes, cellular components, and molecular functions. Most DEGs in the biological process category were enriched in cellular processes, metabolic processes, responses to stimuli, and biological regulation. Most of the DEGs in the cellular component category were enriched in cells, organelles, and membranes. Most of the DEGs in the molecular function category were associated with binding, catalytic activity, and transporter activity (Figure 7C). Some of the DEGs that are associated with processes that are involved in drought stress response were upregulated more than 1.4-fold, such as genes encoding galactinol-sucrose galactosyltransferase 2-like, ABC transporter C family member 9-like, beta-galactosidase 1-like, tyrosine aminotransferase, UDP-glycosyltransferase 76E11-like, calcium-binding protein CML38-like and peroxidase 15-like, and zinc finger CCCH domain-containing protein 20-like. Some were genes encoding proteins that participate in the drought stress response by affecting the ABA pathway, such as ABA 8’-hydroxylase 3, protein phosphatase 2C 37-like, and E3 ubiquitin-protein ligase RGLG1-like (Appendix A). Based on these results, we suggest that GmERF113 plays an important role in the drought stress response by regulating various drought-related genes and may be involved in the ABA-related pathway.

### 2.7. Overexpression of GmERF113 and GmPR10-1 Influences ABA Content and ABA-Related Gene Expression

To further investigate whether the role of GmERF113 in drought tolerance is associated with the ABA pathway, we measured the ABA contents of the WT, *GmERF113*-OE, and *GmERF113*-RNAi plants under normal conditions and drought treatment. We also analyzed the expression of genes that function in drought tolerance and genes that are related to the ABA signaling pathway in WT, *GmERF113*-OE, and *GmERF113*-RNAi plants. The ABA levels were significantly higher in *GmERF113*-OE plants than in the WT and *GmERF113*-RNAi plants under both the drought treatment and normal conditions (Figure 8A). Moreover, the expression level of *GmABA8*’*-OH 3* (encoding ABA 8’-hydroxylase) was markedly higher in *GmERF113*-RNAi plants than in the WT and *GmERF113*-OE plants, and the expression levels of ABA signaling genes *GmRGLG1* and *GmPP2C37* were significantly higher in the *GmERF113*-OE plants than in the WT and *GmERF113*-RNAi plants (Figure 8B).

Finally, to investigate whether *GmPR10-1*, the direct downstream target gene of GmERF113, is also involved in the ABA pathway, we performed the same assay as above using transgenic hairy roots from EV, *GmPR10-1*-OE, and *GmPR10-1*-RNAi plants. The ABA levels were significantly higher in *Gm**PR10-1*-OE than in either EV or *Gm**PR10-1*-RNAi transgenic hairy roots under both drought treatment and normal conditions (Figure 8C). *GmABA8*’*-OH 3* was expressed at markedly lower levels in *Gm**PR10-1*-OE vs. EV hairy roots, whereas *GmRGLG1* and *GmPP2C37* expression were not different in *Gm**PR10-1*-OE vs. EV and *Gm**PR10-1*-RNAi hairy roots (Figure 8D). These results suggest that overexpressing *GmERF113* and *GmPR10-1* increases the ABA contents in soybean plants and hairy roots by affecting the expression of the ABA 8’-hydroxylase gene *GmABA8*’*-OH 3*, thus improving drought resistance.

## 3. Discussion

### 3.1. GmERF113 Contributes to Soybean Response to Drought Stress

Soybean is the most important legume crop globally, representing an important source of edible oils and proteins for human consumption [69]. Drought can cause up to 40–60% losses in soybean production worldwide [70]. Among the DEGs in soybean leaf tissue under drought stress, ERF genes show the most significant differential expression, followed by bHLH, MYB, NAC, and WRKY genes [71]. Moreover, there was some evidence that showed that ERF transcription factors were also involved in regulating drought responses in addition to biotic stresses [72,73,74]. For example, the transgenic Arabidopsis overexpressing *SlERF84* from tomato (*Solanum lycopersicum*) not only showed decreased resistance to the bacterial speck pathogen, *Pseudomonas syringae* pv. *tomato* DC3000, but also displayed increased tolerance to drought stress [74]. Several studies have demonstrated that the ectopic expression of *GmERF3*, *GmERF4*, and *GmERF6* improved drought resistance in tobacco or Arabidopsis and overexpressing *GmERF75* improved the resistance of soybean hairy roots to osmotic stress [60,75,76,77]. Similarly, we have previously demonstrated that *GmERF113* enhanced resistance to *P. sojae* in soybean [61]. Then, we found that *GmERF113* expression was induced by PEG treatment (Figure 1), suggesting that GmERF113 plays a critical role in the drought stress response. Under drought stress, transgenic soybean plants overexpressing *GmERF113* (*GmERF113*-OE) showed significantly slower water loss in the leaves than WT plants and plants with RNAi silencing of the gene (*GmERF113*-RNAi plants) (Figure 2A). The leaves of *GmERF113*-OE plants also exhibited less wilting under drought treatment and stronger recovery after rehydration than those from WT and *GmERF113*-RNAi plants (Figure 2B–D). In addition, *GmERF113*-OE transgenic plants possessed higher SOD activity, POD activity, and proline content (all protective against drought stress), as well as lower MDA content (indicative of drought damage), compared to the WT and *GmERF113*-RNAi plants (Figure 3A–D), providing further evidence that overexpressing *GmERF113* improves plant resistance to drought stress. Collectively, these results demonstrate that GmERF113 contributes to the response of soybean to drought stress.

### 3.2. GmERF113 Increases Soybean Drought Resistance by Directly Activating GmPR10-1 and Affecting the Expression of Drought-Related Genes

Pathogenesis-related (PR) genes, a class of defense-related genes, play important roles in plant resistance to biotic and abiotic stresses [51,56,78,79,80,81]. *PR10* gene expression is induced in many plants by drought and salt stress [48,82,83,84]. Here, we isolated *GmPR10-1* from ‘Dongnong 50’ and produced composite soybean plants with *GmPR10-1* transgenic hairy roots by *Agrobacterium rhizogenes*-mediated transformation. The degree of leaf wilting was lower in the composite plants with *GmPR10-1-*OE transgenic hairy roots compared to the EV control and *GmPR10-1*-RNAi lines under drought treatment. After rehydration, the *GmPR10-1*-OE lines remained viable, whereas the EV and *GmPR10-1*-RNAi lines were dead, with severe leaf shedding and stem shrinkage (Figure 5B–D). In addition, the SOD and POD activity and proline content in the leaves of *GmPR10-1-OE* plants were significantly higher than those of the EV and *GmPR10-1*-RNAi plants, whereas the MDA content showed the opposite trend (Figure 6A–D). Together, these results confirm the role of *GmPR10-1* in drought tolerance in soybean, which is consistent with the finding that some PR10 proteins are involved in drought tolerance [48,54,55,57].

We previously found that GmERF113 binds to the GCC-boxes [61]. (Zhao et al., 2017). Here, we demonstrated that GmERF113 binds directly to the GCC-box in the *GmPR10-1* promoter by ChIP-qPCR and showed that GmERF113 promotes the expression of *GmPR10-1* in a dual luciferase assay. Our RNA-seq results indicated that overexpressing *GmERF113* led to the altered expression of several drought-stress-related genes, including genes encoding galactinol-sucrose galactosyltransferase 2-like, ABC transporter C family member 9-like, beta-galactosidase 1-like, tyrosine aminotransferase, UDP-glycosyltransferase 76E11-like, calcium-binding protein CML38-like and peroxidase 15-like, and zinc finger CCCH domain-containing protein 20-like (Appendix A). Galactitol-sucrose galactosyltransferase 2 interacts with the MYB gene *1R-MYB* in chickpea (*Cicer arietinum* L.), which is significantly induced by drought; these MYB transcription factors are jointly involved in the drought tolerance pathway [85]. The guard cell plasma membrane ABCC-type ABC transporter protein AtMRP4 is involved in regulating stomatal opening and confers drought tolerance in Arabidopsis [86]. Beta-galactosidase and peroxidase function in abiotic stress responses, and beta-galactosidase is a glycosyl hydrolase that is involved in cell wall modification that plays an important role in plant development and adaptation to environmental stress [87,88]. Ectopic expression in *Arabidopsis* of the tyrosine aminotransferase gene *MdTAT2* from apple (*Malus domestica*) or the overexpression of this gene in apple callus enhance tolerance to drought and osmotic stress [89]. Arabidopsis plants overexpressing *AtUGT76C2* (*UDP-glycosyltransferase 76C2*) showed enhanced drought tolerance, and the stress-inducible genes *DREB2A*, *RD22*, *RD29B*, *LEA*, *COR47*, and *KIN1* were significantly upregulated in response to *AtUGT76C2* overexpression [90]. An Arabidopsis calmodulin-like (CML) protein with calcium-binding activity was shown to be involved in regulating ABA signaling and drought stress tolerance in guard cells [91]. The CCCH zinc finger protein OsC3H10 is involved in regulating the drought tolerance pathway by modulating the expression of stress-related genes in rice [92]. These findings suggest that GmERF113 increases drought resistance in soybean by directly activating *GmPR10-1* and affecting the expression of drought-related genes.

### 3.3. GmERF113 Might Participate in ABA-Mediated Regulation of Drought Response

The ERF transcription factor genes *GmERF3*, *GmERF4*, *GmERF5*, *GmERF6*, *GmERF7,* and *GmERF75* are induced by ABA in soybean [35,75,76,77,93]. ABA plays a crucial role in plant development and adaptation to abiotic stress [42,94,95]. Several ABA-related genes regulate stomatal aperture and water potential by affecting the ABA contents to improve the drought resistance of plants [96,97,98]. ABA 8′-hydroxylase is the key enzyme in the oxidative catabolism of ABA. Arabidopsis plants that were treated with Abscinazole-E3M, an inhibitor of ABA 8′-hydroxylase, improved drought resistance [99]. We previously demonstrated that *GmERF113* is induced by ABA [61]. Here, we showed that *GmERF113*-OE plants had significantly smaller stomatal apertures than WT plants under normal conditions (** *P* < 0.01; Figure 2C,D). We propose that GmERF113 negatively regulates the expression of the ABA 8′-hydroxylase gene, *GmABA8*′*-OH 3* (Appendix A, Figure 8B), which in turn leads to significantly higher ABA contents in *GmERF113*-OE plants than in the WT plants, thereby reducing stomatal opening size in *GmERF113*-OE plants (Figure 8A).

In addition, RNA-seq analysis revealed the upregulation of several genes encoding proteins that may be involved in the ABA signaling pathway, such as protein phosphatase 2C 37-like and E3 ubiquitin-protein ligase RGLG1-like (Appendix A). Type 2C protein phosphatases (PP2Cs) negatively regulate ABA signaling and play diverse roles in plant development responses to various stresses [9,100,101]. Arabidopsis ABA promotes the degradation of PP2CA via the E3 ligase RGLG1 E3, and the RING E3 ligase RGLG2 interacts with AtERF53 to negatively regulate the drought stress response in Arabidopsis [45,102]. These findings suggest that GmERF113 functions in ABA signaling by regulating the genes that are related to the ABA signaling pathway.

Finally, in many important plant pathological systems, phytohormones are essential for the activation of signaling during disease defense responses [103,104,105,106,107]. *PR4* and *PR10* were significantly upregulated in both the leaf and stem tissues of highly resistant lentil varieties after treatment with ABA, suggesting that ABA is a key hormonal activator of necrotrophic fungal defense response signaling in lentil [108]. Several *PR10* genes are also induced by ABA [50,109]. Similarly, we found that the stomatal apertures of *GmPR10-1*-OE soybean hairy root composite plants were significantly smaller than those of WT plants under normal conditions (** *P* < 0.01; Figure 5C,D). The ABA content was significantly higher in *PR10-1*-OE transgenic hairy roots than in EV and *PR10-1*-RNAi transgenic hairy roots under both normal and drought treatment, and the expression of *GmABA8*′*-OH 3* was significantly lower in *PR10-1*-OE vs. EV and *PR10-1*-RNAi transgenic hairy roots (Figure 8C,D).

Based on these results, we propose a model that explains the mechanisms of GmERF113 drought stress (Figure 9). When soybean plants are subjected to drought stress, *GmERF113* is rapidly activated and transcribed, and then GmERF113 activates the expression of *PR10-1* by binding to the GCC-box in the *PR10-1* promoter, thereby enhancing the drought resistance of soybean plants. In addition, *Gm*ERF113 promotes the expression of two genes which are in the ABA signaling pathway, *GmPP2C37* and *GmRGLG1*, also both GmERF113 and GmPR10 decrease the expression of an ABA catabolic gene, *GmABA8′-OH 3*, thereby the ABA content in the plant can be increased. The increased ABA level further promotes the closure of stomata and thus improves the drought tolerance of soybean plants.

## 4. Materials and Methods

### 4.1. Plant Materials, Growth Conditions, and Treatments

‘Dongnong 50’, a popular soybean cultivar in Heilongjiang, China, was obtained from the Key Laboratory of Soybean Biology at the Chinese Ministry of Education, Harbin, China, and used for expression analysis, gene transformation experiments, and gene isolation. The seeds were sown in pots containing a vermiculite: soil (1:1) mixture and placed in a growth chamber under a 16 h light/8 h dark photoperiod with 70% relative humidity at 25 °C until the V2 stage [110]. The seedlings were treated with 20% PEG-6000, and the first trifoliate leaves were collected at 0, 1, 3, 6, 9, 12, and 24 h [111]. All the samples were immediately frozen in liquid nitrogen and stored at −80 °C prior to analysis. For the dual-luciferase assays, *Nicotiana benthamiana* seeds were grown in half-strength Murashige and Skoog (1/2MS) medium for 7 days, and the 7-day-old seedlings were transferred to sterile soil and cultured for up to 21 days (vermiculite:soil, 1:1). The plants were maintained at 25 °C under a 16 h light/8 h dark photoperiod.

### 4.2. RNA Isolation and Quantitative Reverse-Transcription PCR

The total RNA was extracted from the samples using TRIzol reagent (Invitrogen, Shanghai, China) following the manufacturer’s protocol. Reverse transcription was conducted using a ReverTra Ace qPCR RT Kit (TOYOBO, Japan). qRT-PCR was employed to measure the gene expression levels using a real-time RT-PCR kit (Toyobo, Osaka, Japan) with a LightCycler 96 System (Roche, Pleasant, CA, USA). The soybean housekeeping gene *GmActin4* (GenBank accession no. AF049106) and *GmTubulin4* (GenBank accession no. XM_003554060) were used as internal reference for normalization of qRT-PCR CT values. The qPCR analyses were performed on three biological replicates, each with three technical replicates. All the qPCR primers that were used in this study are listed in Appendix A.

### 4.3. Construction of the GmERF113 Recombinant Plasmids, Genetic Transformation of Soybean

The *GmERF113*-PMD18T recombinant vector was constructed in our previous study [61]. To generate the 35S:*GmERF113* construct, the full-length coding sequence of *GmERF113* was amplified with specific primers (Appendix A) and inserted into the *Nco*I and *Bgl*II cleavage sites of plant expression vector pCAMBIA3301 with the *Bar* gene and 4× Myc tag as the selectable marker. To obtain the *GmERF113* RNAi vector, a 325-bp fragment of *GmERF113* was amplified using the primer set *ERF1* RNAi-F/R (Appendix A). After the fragment was inserted into the *Bam*HI and *Xba*I sites of the pFGC5941 vector, the reverse repeat fragment was cloned into the *Xho*I and *Nco*I sites of the vector with the *Bar* gene as the selectable marker [112].

The overexpression construct and the RNAi silencing construct were transferred into *Agrobacterium tumefaciens* strain LBA4404 and used to transform the soybean as described by Paz et al. [113]. *GmERF113*-OE transgenic plants (T_3,_ self-crossing) were identified using a QuickStix Kit for Bar (EnviroLogix, Portland, ME, USA), by qPCR and immunoblotting using Myc antibody (Abmart, code number M20002M). *GmERF113-*RNAi T_3_ transgenic plants were tested using bar strips and qPCR.

### 4.4. Drought Treatment, Measuring Water Loss Rate, and Stomatal Aperture

A total of three independent *GmERF113*-overexpression T_3_ seedlings, three independent *GmERF113*-RNAi T_3_ seedlings, and three wild-type (WT) seedlings were grown in the same volume of soil (1:1 vermiculite: soil) and incubated in an incubator at 25 °C under a long daylight cycle (16 h light/8 h dark) and 60% relative humidity until V3 stage [110].

To measure water loss, the second trifoliate leaf was cut from the plant, weighed immediately, and periodically at one-hour intervals following the procedure that was described by Wang et al. [114].

For the drought treatment and the measurement of the stomatal aperture assays, the WT, *GmERF113*-OE, and *GmERF113*-RNAi seedlings were withheld watering for 7 days. The plants were then re-watered for 2 days. The seedling phenotypes were photographed after 0, 5, and 7 days of drought treatment and after 2 days of rehydration using a camera (Canon IXUS 860IS). The second trifoliate leaves of plants were collected at the same time point as mentioned above. The lower epidermis of the leaves was immediately taped and placed under a microscope to observe the morphology of the stomata. The stomata apertures in the images were measured using ViewPoint software. The stomatal aperture results are reflected by the width/length of the stomata.

### 4.5. Measurement of Proline Content, MDA Content, and Superoxide Dismutase and Peroxidase Activities

The second trifoliate leaves of *GmERF113* transgenic and WT soybean plants were collected after 0, 5, and 7 days of drought treatment and after 2 days of rehydration to measure the physiological indexes. Malondialdehyde (MDA) content, proline content, and the superoxide dismutase (SOD) and peroxidase (POD) activities were measured using an MDA Assay kit (Comin, MDA-2-Y, Suzhou, China), Proline Assay kit (Comin, PRO-2-Y, Suzhou, China), SOD Assay kit (Comin, SOD-2-Y, Suzhou, China), and POD Assay kit (Comin, POD-2-Y, Suzhou, China), respectively, according to the manufacturers’ protocols. Each experiment included three replicates, each with three technical replicates.

### 4.6. Chromatin Immunoprecipitation (ChIP) Assay

For the ChIP assays, WT and *p35S*:*GmERF113*-myc transgenic plants were subjected to chromatin extraction and immunoprecipitation as described by Saleh et al. [115]. The second and third trifoliate leaves of the WT and 35S-*GmERF113*-*myc* transgenic plants were harvested at approximately 1 g, respectively. The leaves were fixed in formaldehyde solution for 30 min under a vacuum, and the reaction was terminated by adding 0.15 M of glycine powder. The chromatin complex was isolated and sonicated to about 500 bp fragments. The protein-DNA crosslinks were incubated with anti-Myc-Tag Mouse mAb (Agarose Conjugated) (Abmart, Shanghai, China, code number M20012). The combined DNAs that were eluted, purified, dissolved, and analyzed by quantitative PCR. The ChIP-qPCR results are reported as relative binding units (IP/input). The primers that were used are listed in Appendix A.

### 4.7. Cloning of the GmPR10-1 Promoter and Dual-Luciferase Assay

The *GmPR10-1* promoter sequence was isolated from the genome of ‘Dongnong 50’ and cloned into the pGreenII 0800-LUC vector, which was used as a reporter. The primers *pGmPR10-1F/R* that were used to clone the *GmPR10-1* promoter are listed in Appendix A. *p35S:GmERF113-myc* was used as an effector construct. The effector and reporter constructs were co-transfected into healthy leaves of 21-day-old *N. benthamiana* plants by agroinfiltration. After infiltration, the plants were incubated in the dark for 3 days. The leaves were then sprayed with luciferin (1 mM luciferin and 0.01% Triton X-100) and incubated in the dark for 20 min before being cut off and placed on Chemiluminescence imaging (Tanon 5200) for photographing. Luciferase (LUC) and the Renilla luciferase (REN) activities of the leaf samples were determined using a commercial kit (Promega; PR-E1910). The 35S promoter- driven *Renilla luciferase* (REN) gene in the pGreenII 0800-LUC vector was used as an internal control. The LUC activity was reacted by LUC/REN ratio. The data are the averages of at least three independent replicates, each with three technical replicates.

### 4.8. Agrobacterium Rhizogenes-Mediated Transformation of Soybean Hairy Roots

The full-length cDNA (termed *GmPR10-1*) was isolated from the cDNA of ‘Dongnong 50′ by RT-PCR using primers *GmPR10-1*-F/R (Appendix A). To generate the 35S:*GmPR10-1* construct, the *Bar* gene in the left arm of the plant expression vector pCAMBIA3301 was replaced with the full-length coding sequence of *GmPR10-1* that was amplified with specific primers (Appendix A), and the *GUS* gene was used as a selectable marker. To generate the *GmPR10-1* RNAi vector, a 300-bp fragment of *GmPR10-1* was amplified using the *GmPR10-1* RNAi-F/R primer set (Appendix A) and inserted in forward orientation into the *Bam*HI and *Xba*I sites of pFGC5941 and in reverse orientation into the *Xho*I and *Nco*I sites of this vector. Transgenic soybean hairy roots were generated by *Agrobacterium rhizogenes*-mediated transformation following the methods as described by Kereszt et al. [116].

### 4.9. RNA-Seq Analysis

RNA was extracted from the leaves of three independent 30-day-old *GmERF113*-OE transgenic soybean plants and three WT plants under non-stress conditions using TRIzol reagent and a Spectrum Plant Total RNA Kit (Sigma-Aldrich, St. Louis, MO, USA). Sequencing libraries were generated using a NEB Next Ultra RNA Library Prep Kit for Illumina (NEB, Ipswich, MA, USA) following the manufacturer’s recommendations, and index codes were added to each sample. RNA-seq was performed using the Illumina HiSeq 2500 platform. Each sample generated more than 6 gigabytes of data. The sequencing data were compared with the soybean reference genome (https://phytozome.jgi.doe.gov/pz/portal.html, accessed on 20 December 2019) using TopHat2 [117] to obtain positional information on the reference gene and information about the characteristics of the sequence. Differential expression analysis between the sample groups was performed using DESeq [118] to identify differentially expressed genes (DEGs) between the two biological conditions. During DEG detection, fold change ≥ 1, FDR < 0.05 was used as a screening standard. The functions of the DEGs were annotated by GO enrichment analysis [119]. The experiment was performed on three biological replicates, each with three technical replicates.

### 4.10. Measuring ABA Contents

The ABA contents were measured as described previously [120] using a High Performance Liquid Chromatography kit (COMIN, Suzhou, China) according to the manufacturer’s protocol.

### 4.11. Statistical Analysis

The experiments were performed on three biological replicates, each with three technical replicates, and the results were statistically analyzed using Student’s *t*-test after test for normal distribution with the Shapiro–Wilk tests [121]. IBM SPSS26 package for Windows (IBM, New York, NY, USA) was used for the Shapiro–Wilk tests. A difference was considered to be statistically significant when ** P* < 0.05 or *** P* < 0.01. The bars indicate the standard deviation of the mean.

## 5. Conclusions

In this study, we found that GmERF113 positively regulates the drought response in soybean. We further confirmed that GmERF113 activates the expression of the downstream target gene *GmPR10-1*. The GmERF113-GmPR10-1 pathway improves drought resistance and affects the ABA content in soybean. In addition, we demonstrated that GmERF113 involves the drought response by positively regulating various drought-related genes that are involved in regulating stomatal opening, antioxidant pathways, cell wall modification, and ABA signaling pathways. Our findings indicate the role of GmERF113 in the regulating mechanism of soybean response to drought stress and provide a theoretical basis for the molecular breeding of drought-resistant soybean.

## Figures and Tables

**Figure 1 ijms-23-08159-f001:**
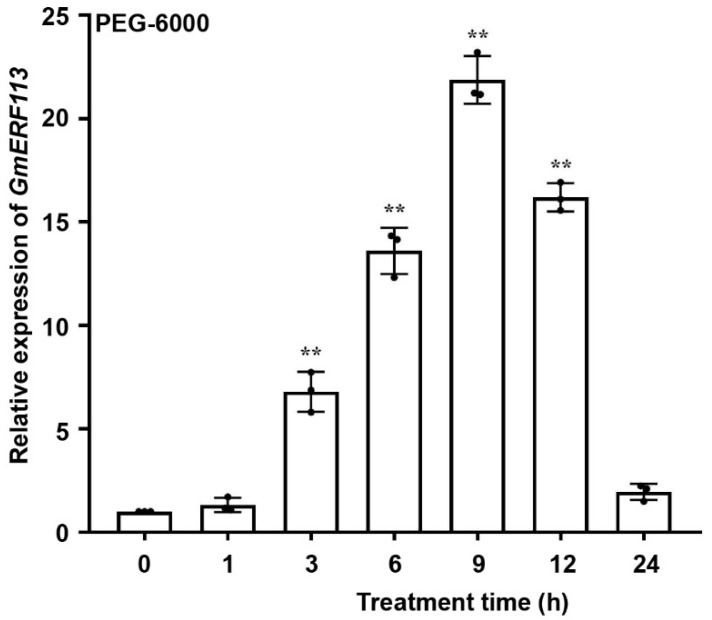
*GmERF113* is induced by drought. Expression patterns of *GmERF113* of soybean seedlings at the V2 stage were examined with 20% PEG-6000 treatment. The first trifoliate leaf samples were collected at 0, 1, 3, 6, 9, 12, and 24 h. The reference soybean gene *GmActin4* (GenBank accession no. AF049106) and *GmTubulin4* (GenBank accession no. XM_003554060) were used as internal controls to normalize the data. The experiment was performed on three biological replicates, each with three technical replicates, and was statistically analyzed using Student’s *t*-test (** *P* < 0.01). The bars indicate the standard deviation of the mean.

**Figure 2 ijms-23-08159-f002:**
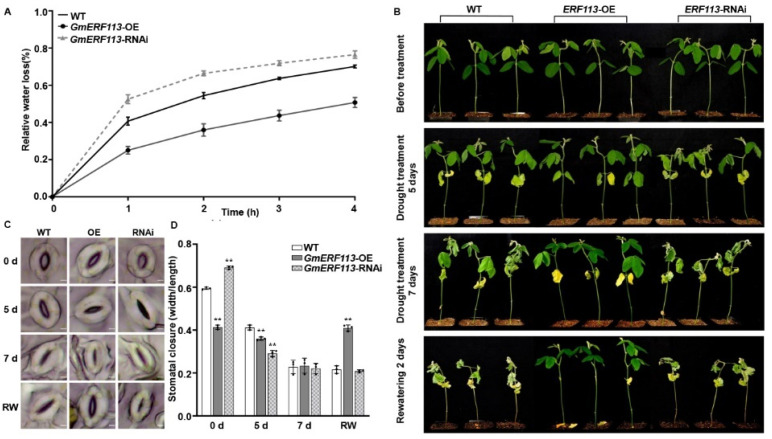
GmERF113 is a positive regulator of plant response to drought. (**A**) The relative water loss of the detached leaves in wild-type (WT) plants, *GmERF113-*OE, and *GmERF113-*RNAi transgenic soybean plants. The second trifoliate leaves of soybean seedlings at the V3 stage were cut which were weighed every hour. The experiment was carried out in three biological replicates, each with three technical replicates. The bars indicate the standard deviation of the mean. (**B**) Phenotypes of WT, *GmERF113-*OE, and *GmERF113-*RNAi soybean plants that were exposed to drought stress for 0, 5, and 7 days and re-watered for 2 days. (**C**,**D**) Stomatal apertures of the second trifoliate leaves of WT, *GmERF113-*OE, and *GmERF113-*RNAi soybean plants that were treated with drought for 0, 5, or 7 d and re-watered for 2 days (**C**), and statistical analysis of the stomatal apertures of each line (bars = 20 μm) (**D**). The experiment was performed on three biological replicates, each with three technical replicates, and was statistically analyzed using Student’s *t*-test (** *P* < 0.01). The bars indicate the standard deviation of the mean.

**Figure 3 ijms-23-08159-f003:**
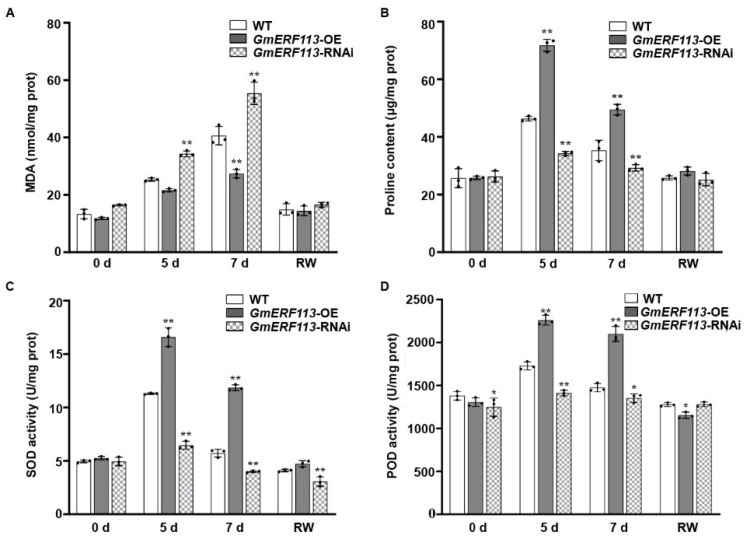
Overexpression or silencing of *GmERF113* alters drought-related parameters that are reflective in soybean plants. (**A**–**D**) Malondialdehyde content (MDA; **A**), proline content (**B**), and superoxide dismutase (SOD; **C**) and peroxidase (POD; **D**) activities of WT, *GmERF113-*OE, and *GmERF113-*RNAi soybean plants that were exposed to drought stress for 0, 5, and 7 days and re-watered for 2 days. The experiment was performed on three biological replicates, each with three technical replicates, and was statistically analyzed using Student’s *t*-test (* *P* < 0.05, ** *P* < 0.01). The bars indicate the standard deviation of the mean.

**Figure 4 ijms-23-08159-f004:**
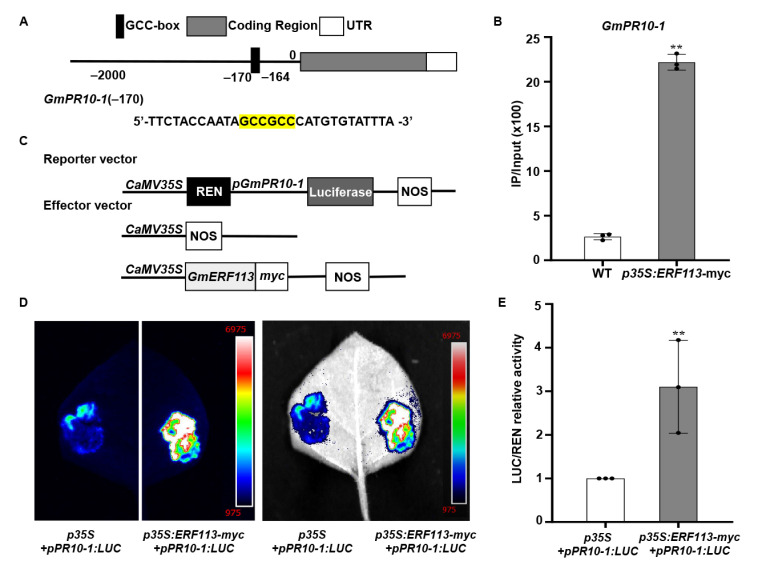
*GmPR10-1* was identified as a target of GmERF113. (**A**,**B**) ChIP analysis of GmERF113 binding to the promoter region of *GmPR10-1*. Chromatin from transgenic soybean plants expressing *Gm**ERF113-myc* or the WT were immunoprecipitated with or without the anti-myc antibody. The precipitated chromatin fragments were analyzed with qPCR using a pair of specific ChIP-qPCR primers, which amplifies the GCC-box (highlighted in yellow) upstream of *GmPR10-1*, as indicated. One-tenth of the input (without antibody precipitation) of chromatin was analyzed and used as a control. A total of three biological replicates, each with three technical replicates, were averaged and statistically analyzed using Student’s *t*-test (*** P* < 0.01). The bars indicate the standard deviation of the mean. (**C**) Schematic representation of the reporter and effector constructs that were used in the dual luciferase assays. (**D**) The dual luciferase assays in tobacco leaves showing that GmERF113 activates the expression of *GmPR10-1* by combining the *GmPR10-1* promoter. Representative pictures were taken. (**E**) LUC/REN activity detection to verify GmERF113 activates the expression of *GmPR10-1*. The combination of the reporter construct (p*GmPR10-1*: LUC) and the blank effector construct [*p35S*] were used as the control. These experiments were performed on three biological replicates, each with three technical replicates, and were statistically analyzed using Student’s *t*-test (** *P* < 0.01). The bars indicate the standard deviation of the mean. “•” were scattered dots which can reflect the distribution of the data.

**Figure 5 ijms-23-08159-f005:**
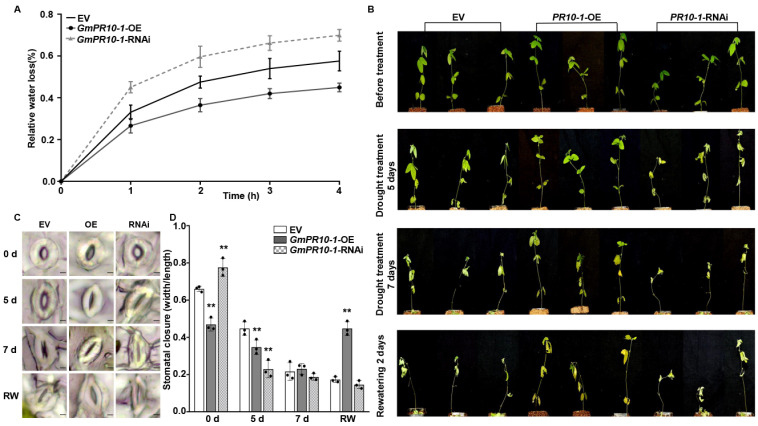
*GmPR10-1* improves the drought tolerance in composite soybean plants with transgenic hairy roots. (**A**) The relative water loss of detached leaves (the second trifoliate leaves from the top) from EV, *GmPR10-1-*OE, and *GmPR10-1-*RNAi composite soybean plants with transgenic hairy roots. Transgenic soybean hairy roots of the same length (~10 cm) were selected, transferred to new pots (soil moisture contents in the pots were kept constant), and incubated for 3 days before the start of the drought treatment. (**B**) Phenotypes of EV, *GmPR10-1-*OE, and *GmPR10-1-*RNAi composite soybean plants with transgenic hairy roots that were exposed to drought stress for 0, 5, and 7 days and re-watered for 2 days. (**C**,**D**) Stomatal aperture of the second trifoliate leaves from the top of EV, *GmPR10-1-*OE, and *GmPR10-1-*RNAi composite soybean plants with transgenic hairy roots that were treated with drought for 0, 5, and 7 days and re-watered for 2 days (bars = 20 μm). (**C**), and statistical analysis of the stomatal apertures of each line. (**D**). The experiments were performed on three biological replicates, each with three technical replicates, and were statistically analyzed using Student’s *t*-test (* *P* < 0.05, ** *P* < 0.01). The bars indicate the standard deviation of the mean.

**Figure 6 ijms-23-08159-f006:**
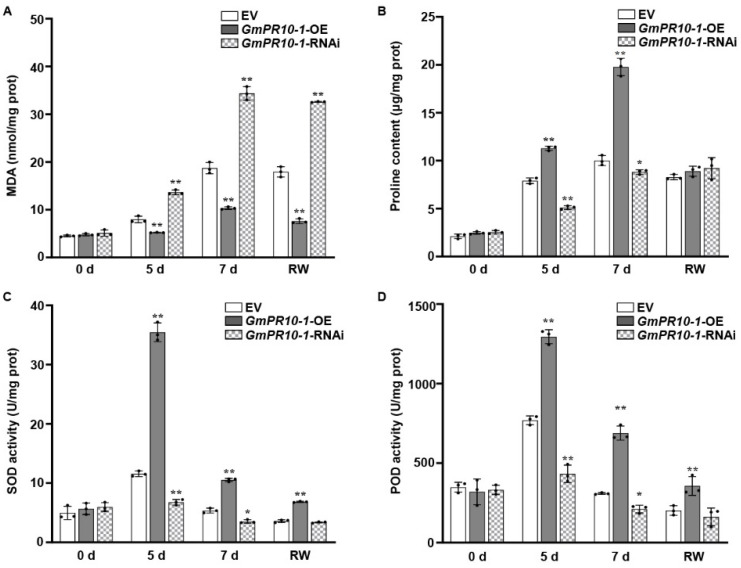
Overexpression or silencing of *GmPR10-1* alters the drought-related parameters in composite soybean plants. (**A–D**) Malondialdehyde content (MDA; **A**), proline content (**B**), and superoxide dismutase (SOD; **C**) and peroxidase (POD; **D**) activities of EV, *GmPR10-1-*OE, and *GmPR10-1-*RNAi composite soybean plants with transgenic hairy roots that were exposed to drought stress for 0, 5, and 7 days and re-watered for 2 days. The second trifoliate from the top was chosen to measure physiological indicators. These experiments were performed on three biological replicates, each with three technical replicates, and were statistically analyzed using Student’s *t*-test (* *P* < 0.05, ** *P* < 0.01). The bars indicate the standard deviation of the mean.

**Figure 7 ijms-23-08159-f007:**
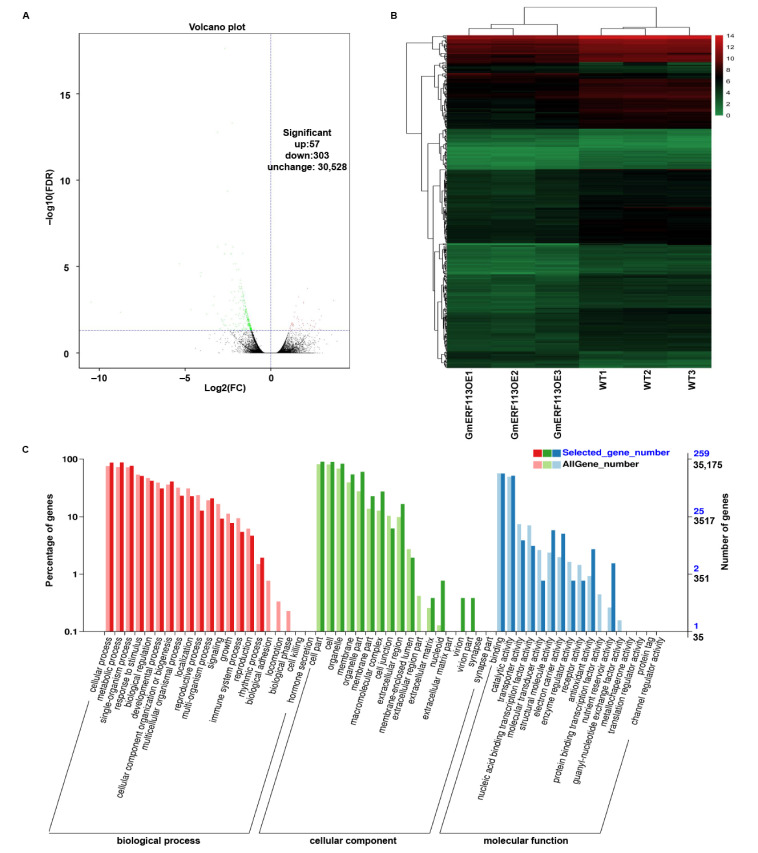
Transcriptomic analysis of gene expression profiles in response to *GmERF113* overexpression. (**A**) Volcano plots of significantly differentially expressed genes in *GmERF113-OE* vs. wild-type soybean plants after the RNA-seq analysis. (**B**) Heat map of significantly differentially expressed genes between the WT and *GmERF113-OE* transgenic soybean plants, as determined using an RNA-seq analysis. Using a false discovery rate < 0.05 and a fold change ≥1 as the screening criteria, a total of 360 differentially expressed genes (DEGs) were identified. The scale bar indicates the fold changes (log_2_ values). (**C**) Gene Ontology functional classification of the differentially expressed genes. The differentially expressed genes were placed into the three main GO categories: biological process, cellular component, and molecular function.

**Figure 8 ijms-23-08159-f008:**
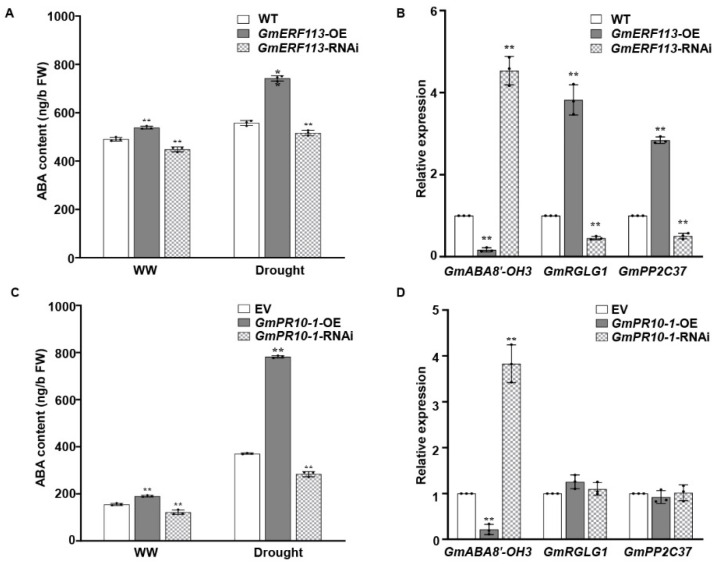
GmERF113 and GmPR10-1 function in ABA responses. (**A**) The ABA contents of *GmERF113* transgenic soybean plants under normal conditions and drought treatment. (**B**) The relative expression of three ABA-related genes that were identified by RNA-seq analysis in *GmERF113* transgenic soybean plants. (**C**) The ABA contents of *GmPR10-1* transgenic soybean hairy roots under normal conditions and drought treatment. (**D**) The relative expression of three ABA-related genes that were identified by RNA-seq analysis in *GmPR10-1* transgenic soybean hairy roots. The reference soybean gene *GmActin4* and *GmTubulin4* were used as an internal control to normalize the data. These experiments were performed on three biological replicates, each with three technical replicates, and were statistically analyzed using Student’s *t*-test (* *P* < 0.05, ** *P* < 0.01). The bars indicate the standard deviation of the mean.

**Figure 9 ijms-23-08159-f009:**
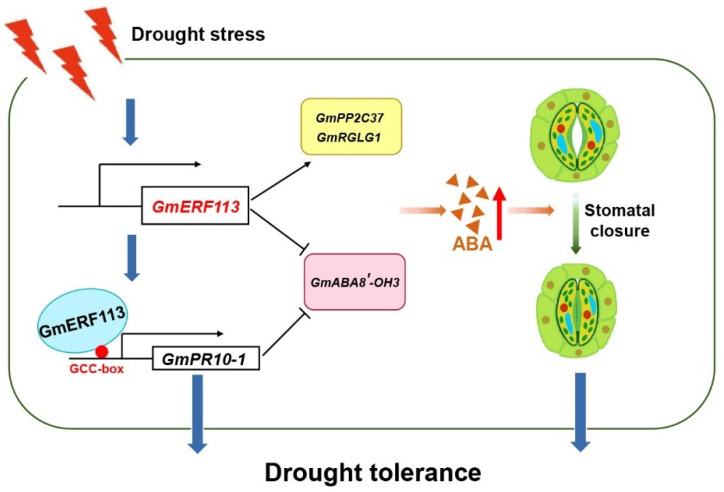
A molecular model of the *GmERF113* in the soybean response to drought stress. When soybean plants are subjected to drought stress, *GmERF113* is rapidly activated and transcribed, and then GmERF113 activates the expression of *PR10-1* by binding to the GCC-box in the *PR10-1* promoter, thereby enhancing the drought resistance of soybean plants. In addition, *Gm*ERF113 promotes the expression of two genes which are in the ABA signaling pathway, *GmPP2C37* and *GmRGLG1*, also both GmERF113 and GmPR10 decrease the expression of an ABA catabolic gene, *GmABA8′-OH 3*, thereby the ABA content in the plants can be increased. The increased ABA level further promotes the closure of stomata and thus improves the drought tolerance of soybean plants.

## Data Availability

The data that support the findings of this study are available from the corresponding author upon reasonable request. Raw RNA sequencing data are available at the NCBI Sequence ReadArchive (SRA) under accession PRJNA760544 and can be accessed via the following link: https://dataview.ncbi.nlm.nih.gov/object/PRJNA760544?reviewer=1jlefrlk5q0np5tv574n095a38, accessed on 5 September 2021. Genes sequences in this research were obtained from National Center for Biotechnology Information (NCBI) (https://www.ncbi.nlm.nih.gov/, accessed on 5 January 2017) and Phytozome (https://phytozome.jgi.doe.gov/, accessed on 16 May 2020). The accession numbers of genes are as follows: *GmERF113* (XM_003548806), *GmPR10-1* (NM_001251335), *GmActin4* (AF049106), *GmTubulin4* (XM_003554060), *Galactinol-sucrose galactosyltransferase 2-like* (Glyma.03G137900), *ABC transporter C family member 9-like* (Glyma.03G101000), *Beta-galactosidase 1-like* (Glyma.08G193500), *Tyrosine aminotransferase* (Glyma.06G235500), *UDP-glycosyltransferase 76E11-like* (Glyma.12G058300), *Calcium-binding protein CML38-like* (Glyma.04G194800), *Peroxidase 15-like* (Glyma.15G128800), *Zinc finger CCCH domain-containing protein 20-like* (Glyma.08G031400), *ABA 8’-hydroxylase 3* (Glyma.16G110700), *Protein phosphatase 2C 37-like* (Glyma.18G035000), *E3 ubiquitin-protein ligase RGLG1-like* (Glyma.06G155100).

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
