# Peer review of "The AP2/ERF GmERF113 Positively Regulates the Drought Response by Activating *GmPR10-1* in Soybean"

_ijms, 2022, doi:10.3390/ijms23158159_

Round 1

Reviewer 1 Report

Manuscript of interest on plant responses to abiotic stress. 

Abstract appropriate to the content of the manuscript.

The introduction provides the most relevant introductory information to the research topic. However, the information at the end of the introduction:  "Here, we performed expression analysis and found that GmERF113 ...  and ABA signaling pathways." should be included in the conclusion. Instead, the authors should clearly state the purpose of the study.

Figure 7C should be enlarged - it is currently unreadable.

Results well discussed and presented in figures form.

Discussion correct.

Statistical analyses were performed correctly using the Student's t-test but please provide information in the M&M on how the normality of the distribution of the variables was found.

Author Response

Reviewer1 Comments to Author:

1.Manuscript of interest on plant responses to abiotic stress.

Response: Thanks for your comment.

2.Abstract appropriate to the content of the manuscript.

Response: Thanks for your comment.

3.The introduction provides the most relevant introductory information to the research topic. However, the information at the end of the introduction:  "Here, we performed expression analysis and found that GmERF113 ...  and ABA signaling pathways." should be included in the conclusion. Instead, the authors should clearly state the purpose of the study.

Response: Thanks for your valuable comments. Following your suggestions, we add the purpose of the study at the end of the introduction as follows: The objectives of this study were (i) to investigate whether GmERF113 is involved in drought stress through expression analysis; (ii) to verify the drought resistance function of GmERF113 through the phenotype and the determination of drought-related physiological indicators; (iii) to resolve a molecular mechanism of GmERF113 in regulating drought resistance in soybean: the GmERF113-GmPR10-1 pathway enhances drought resistance and affects ABA content in soybean, which will help us better understand the drought resistance function of GmERF113 and provide a theoretical basis for molecular breeding of drought-tolerant soybean (Please see page 6 lines 119 to 127 in the revised manuscript).

4.Figure 7C should be enlarged - it is currently unreadable.

Response: As suggested by your valuable advice, we enlarge Figure 7C in the revised manuscript.

5.Results well discussed and presented in figures form.

Response: Thanks for your comments.

6.Discussion correct.

Response: Thanks for your comment.

7.Statistical analyses were performed correctly using the Student's t-test but please provide information in the M&M on how the normality of the distribution of the variables was found.

Response: Thanks for your valuable comment. Following your valuable suggestions, we add the methods applied to test the normal distribution as well as the software tools we used. We add the relevant descriptions in M&M section as follows: and the results were statistically analyzed using Student’s t-test after test for normal distribution with the Shapiro-Wilk tests [121]. IBM SPSS26 package for Windows (IBM, New York, NY, USA) was used for the Shapiro-Wilk tests (Please see page 27 lines 616 to 618); And we add a reference (Please see page 52 lines1149 to 1150).

Reviewer 2 Report

My comments in the attached file

Author Response

1. Write the concentration “Here, we determined that GmERF113 is induced by PEG-4000”.

2. On any basis the authors have selected this concentration? On any basis the authors have selected the tested hours? “The seedlings were treated with 25% PEG-4000, and the first trifoliate leaves were collected at 0, 3, 6, 9, 12, and 24 h”.

Response: Thanks for your valuable comments. Here, we response to the first and second comments together. For the PEG treatment concentration: We chose an intermediate concentration of 25% for the treatment, based on previous reports using 20% and 30% PEG6000. However, we used PEG4000 instead of PEG6000, and in order to make the experiment more reasonable and reliable, we re-performed the experiment using 20% PEG6000, and the results showed the transcript level of GmERF113 increased rapidly from 1 to 9 h (peak level 21.86-fold of the control) and then decreased to a low level by 24 h (Figure 1). For the tested hours we select, we referred to previous reports and selected 0,1, 3, 6, 9, 12, and 24 h. In summary, we have made the following changes: (i) The sentence has been changed to “Here, we determined that GmERF113 is induced by 20%PEG-6000” (Please see page 2 lines 26 to 27); (ii) The sentence has been changed to “the transcript level of GmERF113 increased rapidly from 1 to 9 h (peak level 21.86-fold of the control) and then decreased to a low level by 24 h (Figure 1)” (Please see page 7 lines 152 to 154); (iii) Figure 1 has been changed; (iv) The sentence has been changed to “The seedlings were treated with 20% PEG-6000, and the first trifoliate leaves were collected at 0, 1, 3, 6, 9, 12, and 24 h [111]” (Please see page 21 to 22 lines 485 to 487) and we add a reference (Please see page50 lines 1113 to 1116); (v) The sentence has been changed to “Expression patterns of GmERF113 of soybean seedlings at V2 stage were examined with 20% PEG-6000 treatment” (Please see page 53 lines 1161 to 1162).

3. The authors must check the font, font size.... etc. in the full manuscript.

Response: Thank you for your valuable advice. Following your suggestions, we check and revise the font, size, space etc. in the full manuscript to make it consistent. (i)The space before“Previously” have been checked. (Please see page 2 line 25); (ii)The font size of “were the opposite” have been revised. (Please see page 2 line 32); (iii)The font size and underline of “Pathogenesis-related (PR) genes are known to contribute to plant resistance to biotic stresses...  PR2-, and PR5-overexpressing Arabidopsis plants showed enhanced drought tolerance” have been revised and removed. (Please see page 5 to 6 lines 100 to 113); (iv) The font size and underline of Arachis hypogaea L.;” have been revised and removed. (Please see page 4 line 78)

4. In abstract section, Write the full name before the abbreviation.

Response: Following your valuable advice, we write the full name before the abbreviation of SOD, POD, MDA and ABA: Malondialdehyde (MDA), Superoxide dismutase (SOD), Peroxidase (POD), abscisic acid (ABA). (Please see page 2 lines 30, 31, 39)

5. In introduction section, write the full name for each abbreviation for the first time.

Response: Thank you for your valuable advice. Following your suggestions, we write the full name for ERF for the first time: ethylene response factor (ERF) (Please see page 4 line 71); We check and ensure to write the full name for each abbreviation for the first time in the full manuscript.

6. Write the full name for “WT”.

Response: As suggested by your valuable advice, we write the full name for “WT” for the first time: wide-type (WT). (Please see page 7 line 168); We replace the first “WT” that appears in the figure legends to “wide-type (WT)”. (Please see page53 line 1170)

7. In 3.1 section, Write the full name for “DEGs”.

Response: Following  your valuable advice, we write the full name ‘differential expression genes’ for ‘DEGs’  (Please see page 13 line 307).

8. “Figure 3. Overexpression or silencing of GmERF113 alters drought-related parameters reflective in soybean plants. (A-D) MDA content (A).” Write the full name for MDA.

Response: Following your valuable advice, we change ‘MDA content (A)’ to ‘Malondialdehyde content (MDA; A)’. (Please see page 54 line1187 and page 56 line1230).
